# Can Agricultural Machinery Harvesting Services Reduce Cropland Abandonment? Evidence from Rural China

Ping Xue [1], Xinru Han [1], Yongchun Wang [2] and Xiudong Wang [1,*]

1   Institute of Agricultural Economics and Development, Chinese Academy of Agricultural Sciences, Beijing 100081, China; 82101201276@caas.cn (P.X.); hanxinru@caas.cn (X.H.)
2   Agricultural Information Institute, Chinese Academy of Agricultural Sciences, Beijing 100081, China; wangyongchun@caas.cn
*   Correspondence: wangxiudong@caas.cn; Tel.: +86-10-8210-6163

**Abstract:** Ending hunger, achieving food security, and promoting sustainable agriculture are the main targets of sustainable development goals. It is well known that cropland resources are the most essential factor in achieving sustainable development goals. However, China has been facing the problem of a continuous reduction in cropland resources. Reducing the abandonment of cropland has become an important way to curb the reduction in cropland resources. Can agricultural machinery harvesting services reduce cropland abandonment in rural China? To answer this scientific question, this study employs the Survey for Agriculture and Village Economy data from 8345 samples of 12 provinces in rural China. The extended regression models (i.e., the extended probit regression model and the extended interval regression model) are used to empirically analyze the relationship between agricultural machinery harvesting services accessed by farmers and cropland abandonment. The results are as follows. Agricultural machinery harvesting services accessed by farmers significantly reduced the probability of cropland abandonment and the proportion of the area of abandoned cropland in farmers' contracted cropland area decreased by 18.5% and 20.3%, respectively. Moreover, the heterogeneity analysis results showed that farmers' access to agricultural machinery harvesting services significantly reduced cropland abandonment in small-scale groups, without elderly households, with nonagricultural income groups, and in the eastern region. This study also provides some policy implications for policymakers to reduce cropland abandonment in rural China.

**Keywords:** cropland abandonment; agricultural machinery harvesting services; extended regression models; rural China

## 1. Introduction

Ending hunger, achieving food security, and promoting sustainable agriculture are the main targets of sustainable development goals worldwide [1]. In this context, how to use limited cropland resources to feed a large population has become a key issue for the present and future in China, even though China has fed 20% of the world's population with approximately 7% of the world's cropland [2].

Cropland resources are the most essential factor for ensuring food security and promoting the development of sustainable agriculture [3,4]. According to the statistics of FAO in 2019, China's cropland area ranks third in the world, which is only lower than that of India and the United States. In terms of the per capita cropland area, however, China is only 0.09 hectares, lower than the 0.12 hectares in India and 0.48 hectares in the United States. This evidence shows that China still faces the problem of insufficient cropland resources. In addition, with the acceleration of urbanization and industrialization, the continuous reduction in cropland resources has further exacerbated this problem [3,5]. Previous studies mainly put forward two ways to curb the reduction of cropland resources: improving the existing cropland use efficiency, and reducing cropland abandonment [6–8]. Admittedly,

improving the use efficiency of cropland alone is not enough to deal with cropland resource reduction, which should also reduce cropland abandonment.

It is well known that cropland abandonment has already become a common phenomenon in the world, which includes both developed countries (e.g., the United States, Australia, and Japan) and developing countries (e.g., China, Chile, Latin America, and Southeast Asia) [9–12]. It has become an increasingly important issue in China since 2000. For example, China's government issued the "Urgent Notice on Resuming the Production of Abandoned Cropland as soon as possible" in 2004, indicating that cropland has been abandoned to varying degrees in some areas. Moreover, large-scale cropland abandonment has occurred in China since 2005, especially in the mountainous counties [11]. This has posed challenges and threats to China's food security and sustainable agriculture [5,13,14]. It has also caused a series of environmental issues, such as the loss of agro-biodiversity and species richness, soil erosion, shallow landslides, and desertification [15–18]. In this context, the Chinese government has paid great attention to this issue and promulgated a series of policies. For example, the "Guiding Opinions on the Overall Utilization of Abandoned Cropland to Promote the Development of Agricultural Production" emphasized the importance and urgency of curbing the abandonment of cropland. In addition, it pointed out that one means of alleviating the abandonment of cropland is by cultivating agricultural professional service organizations to provide services for migrant workers and farmers with weak labor ability [19]. This also provides some inspiration for our study.

Much research has been done on the reasons for cropland abandonment. On the one hand, some studies indicate that rural laborers' migration to cities is the main factor that leads to cropland abandonment, such as Xu et al. (2018) [10] and Gao et al. (2020) [20]. In particular, with the arrival of the Lewis turning point in rural China in 2003, the era of the unlimited supply of rural labor force has passed [21]. China's unique household contract responsibility system, that is, that farmers have only the right to use the land but not the right to sell, has restricted farmers from selling cropland. In this case, the migrant workers can only transfer out of their cropland, therefore, the part of the cropland that cannot be transferred out of, will be abandoned. On the other hand, high agricultural production costs, such as the high investment costs of agricultural machinery, are also the main factor leading to cropland abandonment [22]. In addition, the croplands that are located far away from the villages and towns may be abandoned [23–25]. To sum up, the low agricultural production capacity, due to the lack of an agricultural labor force and operation equipment, is the main factor leading to the abandonment of cropland.

In terms of the driving mechanism for reducing cropland abandonment, previous research mainly explored land transfer [7,26], population aging [27], agricultural cooperatives [28], internet use [29], etc. Few studies have focused on the critical factors (i.e., agricultural production capacity) in the reduction of cropland abandonment. Agricultural mechanization services may be an effective way of improving agricultural production capacity, which is a special form of helping farmers to achieve the mechanized operation of part or all of the agricultural production links in rural China. It can alleviate rural labor shortages, reduce agricultural production costs, and improve agricultural mechanization levels [30–33]. This may effectively reduce cropland abandonment. In particular, harvesting is the most time-consuming and labor-intensive step in agricultural production (i.e., the "heaviest" of the agricultural production links) [30,34], which is more likely to lead farmers to abandon cropland. Therefore, this paper mainly focuses on agricultural machinery harvesting services (AMHSs).

In summary, the main aim of this paper was to explore whether AMHSs accessed by farmers can reduce cropland abandonment. To achieve this aim, we used the data of the Survey for Agriculture and Village Economy (SAVE) in 2019 and 2020 and employed the extended regression models (ERMs). Precisely, the main questions answered in this study are as follows: Can AMHSs reduce cropland abandonment in rural China? What is the heterogeneity in the impact of AMHSs on cropland abandonment?

Compared with the previous studies, there are mainly three marginal contributions of our study. First, different to previous quantitative studies, which mainly focus on the reasons for cropland abandonment, such as Deng et al. (2018) [27], Xu et al. (2018) [10], etc., the main aim of our study was to qualitatively explore the factors for reducing cropland abandonment. Second, we employed the extended regression models (i.e., the extended probit regression model and the extended interval regression model) to circumvent the endogenous problems caused by the reverse causality between AMHSs access and cropland abandonment and the problem of self-selection. Moreover, compared with IV-Probit or IV-Tobit, this model is suitable for binary endogenous explanatory variables. Third, this is the first study, to the best of our knowledge, to analyze whether access to AMHSs can reduce cropland abandonment in rural China.

## 2. Methods and Data

### 2.1. Theoretical Framework

In this section, we constructed a theoretical framework to analyze the relationship between access to AMHSs and cropland abandonment. As rational economic men, farmers maximize their income mainly through the rational allocation of labor and land resources [35]. With the growth of off-farm wages, farmers tend to allocate more labor resources to the nonagricultural sectors, which will lead to the reduction of labor input in agricultural production [24,36]. This phenomenon has induced serious cropland abandonment in rural China, that is, most of their cropland has been abandoned due to the lack of a sufficient labor force to manage it [10,13]. Although land transfers can alleviate the cropland abandonment to a certain extent, the rural land transfer market is still imperfect and the land transfer degree is still low [37]. Moreover, most of the migrant workers cannot get social security in cities, and most of the farmers still have a "land complex", so they would rather abandon the cropland than transfer it out [11,38,39]. In addition, the expensive input of agricultural machinery also prevents farmers from investing in machinery to replace labor input for agricultural production [40].

The agricultural mechanization services may provide a feasible approach to reducing cropland abandonment caused by the above dilemmas, especially in AMHSs. Figure 1 shows the theoretical framework for the impact of agricultural mechanization services access on cropland abandonment.

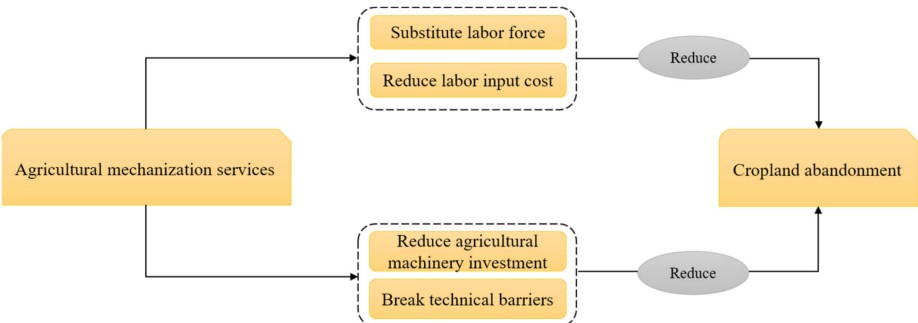

**Figure 1.** The figure of theoretical framework.

On the one hand, it can effectively substitute for the labor force in agricultural production, even if the land managed by the farmers is small and scattered [31,33]. In this case, the impact of labor migration on cropland abandonment will be weakened. In addition, with the increase in nonagricultural wages and income, the relative costs of agricultural labor input are high, which makes farmers reduce labor input in agricultural production. Previous studies proved that the cost of agricultural mechanization services is relatively lower than that of agricultural labor input, especially in the labor-intensive production links (such as the harvesting links) [41,42]. This will prevent farmers from abandoning their cropland due to high labor input costs in agricultural production.

On the other hand, purchasing agricultural machinery is expensive for most farmers, especially for the low-income level groups [43], which leads farmers to give up their cropland and leave agricultural production, due to a lack of agricultural machinery. Moreover, there are high technical barriers for most farmers to use agricultural machinery [44]. Therefore, farmers' access to agricultural mechanization services may be a better way to carry out agricultural production, which can alleviate the impact of the low level of agricultural mechanization on cropland abandonment. To sum up, agricultural mechanization services (mainly AMHSs in this study) may reduce cropland abandonment, which still needs to be tested by subsequent empirical analysis.

Accordingly, we mainly propose the following two hypotheses:

**Hypothesis 1.** *AMHSs accessed by farmers can reduce cropland abandonment in rural China.*

**Hypothesis 2.** *AMHSs can effectively alleviate the impact of labor migration on cropland abandonment.*

### 2.2. Study Methods

The extended regression models (ERMs) were employed in this study to evaluate the impact of AMHSs accessed by farmers on cropland abandonment. On the one hand, one dependent variable is binary (i.e., whether farmers abandon cropland), and another dependent variable (the proportion of the area of abandoned cropland in farmers' contracted cropland area) is a truncated variable in this study. On the other hand, the key variable, whether farmers access AMHSs, is a binary that may have a reverse causality with the dependent variables. In addition, the AMHSs accessed by farmers is a self-selection process, which will produce selection bias due to unobserved factors of farmers (such as agricultural management ability and the ability to accept new things). This can lead to endogeneity problems that make the estimated results biased. In this context, previous studies mainly adopted the IV-Probit and IV-Tobit model for a binary dependent variable and a truncated dependent variable, respectively. The above models only fit continuous endogenous covariables [45], while the key endogenous variable (i.e., whether farmers access AMHSs) is binary in our study. Therefore, we adopted the extended regression models (ERMs), which can fit the binary endogenous covariables. In particular, we adopted an extended probit regression for the binary dependent variable and an extended interval regression for the truncated dependent variable. These two benchmark models are given as:

$$LA_i = \alpha_0 + \alpha_1 AMHS_i + \alpha_2 X_i + \alpha_3 Year + \alpha_4 Region + \mu_i \qquad (1)$$

$$LA_{ip} = \beta_0 + \beta_1 AMHS_i + \beta_2 X_i + \beta_3 Year + \beta_4 Region + \mu_i \qquad (2)$$

where $LA_i$ and $LA_{ip}$ represent whether farmers $i$ abandon cropland and the proportion of the area of abandoned cropland in the contracted cropland area of farmers $i$, respectively; $AMHS_i$ represents whether farmers $i$ access $AMHSs$; $X_i$ are the vectors of other control variables; $Year$ and $Region$ represent dummy variables of year and provinces, respectively; $\alpha_0$–$\alpha_4$ and $\beta_0$–$\beta_4$ are the vectors of the parameters; and $\mu_i$ and $\mu_i$ are the error terms.

This study also introduced an instrumental variable to circumvent the endogeneity problem. Following Kung (2002) [46] and Deng et al. (2018) [7] etc., this study selected the percentage of other farmers in the same village who access AMHSs as an instrumental variable. On the one hand, this instrumental variable, related to endogenous covariables, is satisfied, that is, the percentage of the other farmers in the same village who access AMHSs directly affects the probability that the focal farmer accesses AMHSs. On the other hand, it should not be related to the dependent variables, i.e., the percentage of other farmers in the same village who access AMHSs does not directly affect the focal farmer's abandoned

cropland. Thus, this instrumental variable is reasonable for our study. The instrumental variable is calculated as follows:

$$IV\_AHMS_{ni_l} = (\sum_{i \neq i_l}^{j} AHMS_i)/(j-1) \tag{3}$$

where $IV\_AHMS_{ni_l}$ represents the probability of access to AMHSs by other farmers in the village $n$ except for farmers $i_l$; $j$ represents the number of samples surveyed in village $n$.

*2.3. Data Source*

This study used micro-level data from the 2019 and 2020 Survey for Agriculture and Village Economy (SAVE) [47,48]. This survey is launched and conducted annually by the Institute of Agricultural Economics and Development (IAED) at the Chinese Academy of Agricultural Sciences (CAAS). It covers 37 counties, 65 towns, and 292 villages in 12 provinces of Hebei, Jilin, Heilongjiang, Zhejiang, Anhui, Fujian, Henan, Hunan, Sichuan, Yunnan, Shaanxi, and Xinjiang (Figure 2). Moreover, it also includes surveys of rural households and villages, which provide abundant information about the rural households, household income, land use, villages, etc.

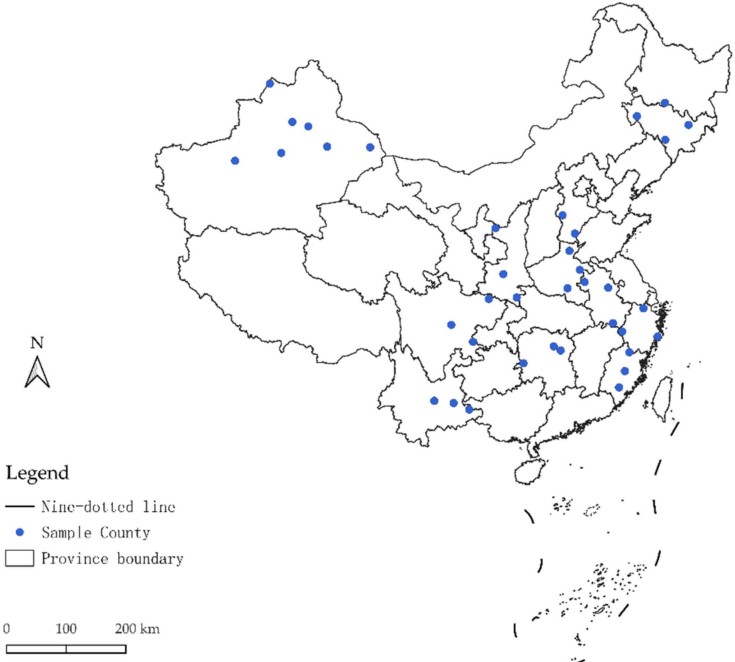

**Figure 2.** The geographical location of the study areas.

To accurately analyze the relationship between AMHSs' access and cropland abandonment, this study processed the data as follows: (1) The national fixed base Consumer Price Index (CPI, 2012 = 100) was used to process income-related variables to eliminate the impact of inflation; (2) This study mainly focused on the impact of the AMHSs accessed by the cropland actual operators on the abandonment of cropland, so the samples with zero actual cropland area are deleted. Finally, 8345 samples were used in this study, which includes 4518 samples from 2019 and 3827 samples from 2020. As mentioned above, unbalanced panel data were used in this study.

*2.4. Definition of the Model Variables*

This study focuses on the impacts of AMHSs accessed by farmers on cropland abandonment in rural China. To achieve this goal, we defined the dependent variables, key variables, and other control variables, as follows.

Following Xu et al. (2018) [10], Deng et al. (2018) [27], this study defined cropland abandonment through the behavior and degree of abandoned cropland as dependent variables. Namely, the behavior of abandoned cropland refers to whether farmers abandoned cropland. The degree of abandoned cropland refers to the proportion of the area of abandoned cropland in farmers' contracted cropland area. The key variable was defined by the AMHSs accessed by farmers, i.e., whether farmers accessed AMHSs in agricultural production. The theoretical analysis above shows that labor migration is the main reason for cropland abandonment, so this study also defined the proportion of nonagricultural income in total income as a key variable. In addition, this study also defined other control variables that may affect the dependent variables. According to the previous research related to the driving mechanisms of cropland abandonment (e.g., Wang et al. (2022) [26], Ma et al. (2020) [28]), this study defined the control variables as the characteristics of the household head (e.g., gender, age, years of education, village cadre status, multiple occupations, and internet access with mobile phone), the household (i.e., the proportion of children, the proportion of seniors, and access to credit), the agricultural production (i.e., the area of cropland, the number of land blocks, agricultural machinery ownership, transfers of land out, purchase of agricultural production insurance, and participation in a cooperative), and the village characteristics (i.e., the disputes relating to contracted land, location). In addition, dummy variables for the year and provinces were also included in this study.

## 3. Results and Analysis

### 3.1. Descriptive Statistics Analysis

The descriptive statistics analysis results are given in Table 1. For cropland abandonment, a total of 14.58 percent of farmers chose to abandon their cropland, and the average proportion of the area of abandoned cropland in farmers' contracted cropland area was 8.46 percent. These results were in accordance with China's actual situation, that most farmers do not abandon their cropland [7,26]. The average proportion of farmers who accessed AMHSs was 29.81 percent. For the characteristics of the household head, most of the household heads were male, the average age was 54.10, and the average years of education were 7.58. The proportion of household heads who were village cadre was 12.13 percent, who engaged in multiple occupations was 35.18 percent, and who accessed the internet with a mobile phone was 42.62 percent. For the characteristics of the household, the average proportion of children and seniors was 11.91 percent and 13.69 percent, respectively. A total of 13.11 percent of households had accessed credit. The proportion of nonagricultural income in the total household income was 60.26 percent, which also suggested that nonagricultural income has become an important part of farmers' income, with the rapid urbanization and industrialization. For the characteristics of agricultural production, the average area of cropland was 1.14 hectares and the average number of land blocks was 4.59. A total of 43.12 percent of households owned agricultural machinery, 47.69 percent of households transferred land out, 30.90 percent of households had purchased agricultural production insurance, and 11.37 percent of households participated in a cooperative. For the characteristics of villages, 36.83 percent of the villages had disputes relating to contracted land, and 13.16 percent of the villages were now located in the town.

**Table 1.** Descriptive statistics analysis results.

| Variables | Description | Mean | SD |
|---|---|---|---|
| Cropland abandonment | Whether farmers abandoned cropland (1 = Yes; 0 = No) | 0.15 | 0.35 |
| The proportion of cropland abandonment | The proportion of the area of abandoned cropland in farmers' contracted cropland area (%) | 8.46 | 25.41 |
| AMHSs access | Whether household accesses AMHSs (1 = Yes; 0 = No) | 0.30 | 0.46 |
| Gender | Gender of household head (1 = Male; 0 = Female) | 0.94 | 0.24 |
| Age | Age of household head (Years) | 54.10 | 10.31 |
| Education | Years of education of household head (Years) | 7.58 | 3.03 |
| Village cadre status | Whether the household head is a village cadre (1 = Yes; 0 = No) | 0.12 | 0.33 |
| Multiple occupations | Whether household head engaged in multiple occupations (1 = Yes; 0 = No) | 0.35 | 0.48 |
| Internet access | Whether household head accesses the internet with mobile phone (1 = Yes; 0 = No) | 0.43 | 0.49 |
| Proportion of children | The proportion of children under the age of 14 (%) | 11.91 | 16.13 |
| Proportion of seniors | The proportion of seniors over the age of 65 (%) | 13.69 | 25.82 |
| Access to credit | Whether household has access to credit (1 = Yes; 0 = No) | 0.13 | 0.34 |
| Proportion of nonagricultural income | The proportion of nonagricultural income in the total household income (%) | 60.26 | 36.12 |
| Area of cropland | Area of cropland of household management (ha) | 1.14 | 1.92 |
| Land blocks | Number of land blocks | 4.59 | 4.29 |
| Agricultural machinery ownership | Whether household owns agricultural machinery (1 = Yes; 0 = No) | 0.43 | 0.50 |
| Transfers of land out | Whether household transfers land out (1 = Yes; 0 = No) | 0.48 | 0.50 |
| Agricultural production insurance | Whether household purchases agricultural production insurance (1 = Yes; 0 = No) | 0.31 | 0.46 |
| Cooperative participation | Whether household participates in a cooperative (1 = Yes; 0 = No) | 0.11 | 0.32 |
| Contracted land dispute | Whether contracted land disputes occur in the village (1 = Yes; 0 = No) | 0.37 | 0.48 |
| Village location | Whether the village is located in the town (1 = Yes; 0 = No) | 0.13 | 0.34 |
| Observations | | 8345 | |

### 3.2. The Impacts of AMHSs Access on Cropland Abandonment

We employed an extended probit regression and an extended interval regression to empirically analyze the impacts of AMHSs access on the behavior and degree of cropland abandonment. The identification strategy of adding control variables step by step was used in these models, where the first regression only controlled the dummy variables of year and provinces, and the second regression added other control variables based on the first regression. The results are shown in Table 2; Model 1 and Model 2 show the extended probit regression model estimation results for whether farmers abandoned their cropland; and Model 3 and Model 4 show extended interval regression model estimation results for the proportion of cropland abandoned by farmers. According to these models, the results of the endogenous test (i.e., H0: endogenous variables are independent of the dependent variables) are all significant at the level of 5%, which indicates that the endogenous variables are related to the dependent variables, and it is appropriate to add the instrumental variable to these models. The results of AMHSs access significantly reduce the cropland abandonment, and Hypothesis 1 was verified. This is in accordance with Deng et al. (2018) [7], who indicated that agricultural mechanization services can alleviate the abandonment of cropland. Based on this, the interpretation of the model results is mainly based on Model 2 and Model 4.

**Table 2.** The estimation results of cropland abandonment.

| Variables | Cropland Abandonment | | The Proportion of Cropland Abandonment | |
|---|---|---|---|---|
| | **Model 1** | **Model 2** | **Model 3** | **Model 4** |
| AMHSs access | −0.202 *** | −0.185 ** | −0.244 ** | −0.203 * |
| | (0.068) | (0.073) | (0.109) | (0.115) |
| Gender | | 0.103 | | 0.138 |
| | | (0.077) | | (0.119) |
| Age | | 0.002 | | 0.001 |
| | | (0.002) | | (0.003) |
| Education | | −0.017 *** | | −0.022 ** |
| | | (0.007) | | (0.010) |
| Village cadre status | | −0.070 | | −0.103 |
| | | (0.057) | | (0.088) |
| Multiple occupations | | 0.015 | | −0.028 |
| | | (0.040) | | (0.062) |
| Internet access | | 0.161 *** | | 0.262 *** |
| | | (0.039) | | (0.061) |
| Proportion of children | | 0.132 | | 0.186 |
| | | (0.117) | | (0.180) |
| Proportion of seniors | | −0.136 | | −0.255 * |
| | | (0.084) | | (0.130) |
| Access to credit | | 0.021 | | 0.007 |
| | | (0.057) | | (0.089) |
| Proportion of nonagricultural income | | 0.111 * | | 0.231 ** |
| | | (0.063) | | (0.097) |
| Area of cropland | | −0.054 *** | | −0.114 *** |
| | | (0.016) | | (0.027) |
| Land blocks | | 0.026 *** | | 0.039 *** |
| | | (0.005) | | (0.007) |
| Agricultural machinery ownership | | 0.025 | | −0.022 |
| | | (0.043) | | (0.065) |
| Transfers of land out | | 0.195 *** | | 0.088 |
| | | (0.038) | | (0.059) |
| Agricultural production insurance | | −0.019 | | −0.034 |
| | | (0.043) | | (0.067) |
| Cooperative participation | | −0.014 | | −0.060 |
| | | (0.060) | | (0.093) |
| Contracted land dispute | | 0.011 | | 0.108 * |
| | | (0.040) | | (0.061) |
| Village location | | 0.219 *** | | 0.497 *** |
| | | (0.053) | | (0.082) |
| Year dummy | Yes | Yes | Yes | Yes |
| Province dummies | Yes | Yes | Yes | Yes |
| Constant | −1.829 *** | −2.063 *** | −2.963 *** | −3.078 *** |
| | (0.096) | (0.193) | (0.183) | (0.316) |
| Instrumental variable | Yes | Yes | Yes | Yes |
| Endogenous test | 0.143 *** | 0.143 *** | 0.107 ** | 0.104 ** |
| | (0.047) | (0.050) | (0.047) | (0.050) |
| Wald χ2 | 523.02 *** | 613.00 *** | 363.88 *** | 405.97 *** |
| Observations | 8345 | 8345 | 8345 | 8345 |

*** $p < 0.01$, ** $p < 0.05$ and * $p < 0.1$.

As shown in Model 2, the AMHSs accessed by farmers reduced the cropland abandonment at the 5% statistical significance level and the probability of farmers reducing cropland abandonment was 18.5%. In terms of the impact of other control variables on farmers' cropland abandonment, the years of education of household heads significantly reduced the abandonment of cropland. The household heads' access to the internet with a mobile phone can significantly increase cropland abandonment, because it can promote farmers participation in off-farm work through convenient access to employment informa-

tion [20,49,50]. Moreover, the proportion of nonagricultural income in the total household income also has a significant and positive impact on cropland abandonment, which also proved that off-farm employment is a major factor leading to cropland abandonment [10]. In addition, the land status, such as the area of cropland, the number of land blocks, and the land transfers, is also an important determinant of cropland abandonment. Specially, the larger the area of cropland managed by farmers, the lower the probability of abandoning their cropland. This is consistent with Yan et al. (2016) [51], who suggested that expanding the scale of cropland management is an effective way of reducing the abandonment of cropland. However, the number of the land blocks has significantly increased the cropland abandonment. Zhang et al. (2014) [25] also proved that much-fragmented cropland has been abandoned in rural China. Farmers' transfers of land out also significantly increased the probability of cropland abandonment. In terms of the characteristics of the village, farmers tended to abandon their cropland when their village was now located in the town, which also proved that the process of urbanization has accelerated the abandonment of cropland [52].

As shown in Model 4, the AMHSs accessed by farmers significantly reduced the proportion of the area of abandoned cropland in farmers' contracted cropland area, and farmers' access to AMHSs can reduce the proportion of cropland abandonment by 20.3%. This study will not detail all of the regression results here to save space. It is worth noting that the proportion of seniors has a significant and negative impact on cropland abandonment. This is in line with Deng et al. (2018) [27], who indicated that elderly farmers help curb cropland abandonment. Many migrant workers do not want to give up their rural land use rights to maintain social security and benefits [38,53], so they may give the cropland to the elderly farmers in the household for management. In addition, the villages with disputes relating to contracted land significantly increased the proportion of abandoned cropland, which also suggested the importance of stable use rights of cropland.

*3.3. Robustness Check*

In this section, we tested the robustness of the estimation results. First, following Xu et al. (2018) [10], we used a Probit model and a Tobit model to test the robustness of the results estimated by an extended probit regression and an extended interval regression, respectively. As shown in Table 3, AMHSs access reduced cropland abandonment in both the Probit and Tobit models but the results were not significant. This indicates that estimation results are biased in the above two models, due to ignoring the endogenous problems. Second, we also compared the results estimated by the IV-Probit model and the IV-Tobit model. The results are given in Table 3; the exogenous Wald test values are both significantly non-zero at the 5% statistical level, rejecting the hypothesis that all of the explanatory variables are exogenous. Moreover, the access to AMHSs has significantly reduced the cropland abandonment in the above two models, while the coefficients estimated by the IV-Probit model were bigger than the extended probit regression model and estimated by the IV-Tobit model were smaller than the extended interval regression model.

In addition, we also compared the results estimated by different key variables to test the robustness [29]. Our study selected the variable of whether farmers' access to the machinery plowing, sowing, and harvesting services (i.e., comprehensive mechanized services (CMSs)) to approximately replace the original key variable. In particular, machinery plowing, sowing, and harvesting are the main links of agricultural production. The farmers' access to the above three links of services can represent their agricultural production capacity. Moreover, AMHSs are an important part of CMSs. As shown in Table 3, the results of the endogenous test are both significant at the level of 1%, which also proved that the endogenous variables are related to the dependent variables. CMSs' access significantly reduced the probability of cropland abandonment and the proportion of the area of abandoned cropland in farmers' contracted cropland area.

**Table 3.** Robustness check models estimation results.

| Variables | Cropland Abandonment | | | The Proportion of Cropland Abandonment | | |
|---|---|---|---|---|---|---|
| | **Probit Model** | **IV-Probit Model** | **Different Key Variable [a]** | **Tobit Model** | **IV-Tobit Model** | **Different Key Variable [b]** |
| AMHSs access | −0.014 (0.043) | −0.199 ** (0.078) | - - | −0.005 (0.040) | −0.132 * (0.072) | - - |
| CMSs access | - | - | −0.381 *** (0.083) | - | - | −0.423 *** (0.132) |
| Control variables | Yes | Yes | Yes | Yes | Yes | Yes |
| Year dummy | Yes | Yes | Yes | Yes | Yes | Yes |
| Province dummies | Yes | Yes | Yes | Yes | Yes | Yes |
| Constant | −2.099 *** (0.193) | −2.060 *** (0.193) | −1.995 *** (0.193) | −1.878 *** (0.185) | −1.858 *** (0.185) | −3.015 *** (0.316) |
| Instrumental variable | No | Yes | Yes | No | Yes | Yes |
| Wald test of exogeneity | - | 8.14 *** | - | - | 4.48 ** | - |
| Endogenous test | - | - | 0.274 *** | - | - | 0.210 *** |
| Observations | 8345 | 8345 | 8345 | 8345 | 8345 | 8345 |

*** $p < 0.01$, ** $p < 0.05$ and * $p < 0.1$; [a] the results estimated by the extended probit regression model; [b] the results estimated by the extended interval regression model.

In summary, the above results confirmed that the results estimated by the extended probit regression model and the extended interval regression model are robust. In addition, this also proved that the IV-Probit model and IV-Tobit model cannot fit the binary endogenous covariables well.

*3.4. Heterogeneity Analysis*

According to the results of Table 2, the control variables of the area of cropland under household management, the proportion of seniors, and the proportion of nonagricultural income all have a significant impact on cropland abandonment. Thus, this study further analyzed the heterogeneity of the impact of AMHSs access on cropland abandonment across different scales of cropland managed by farmers, the household composition (i.e., whether this was a household with seniors), and nonagricultural income (i.e., with and without nonagricultural income). Furthermore, we also analyzed the heterogeneity across different regions (i.e., the eastern region includes the provinces of Hebei, Zhejiang, and Fujian, the central region includes the provinces of Jilin, Heilongjiang, Anhui, Henan, and Hunan, and the western region includes the provinces of Sichuan, Yunan, Shaanxi, and Xinjiang). To save space, this study only listed the results of the impact of AMHSs access on whether farmers abandon their cropland. The results are shown in Table 4.

For the different cropland scales, AMHSs accessed by farmers significantly reduced cropland abandonment only in small-scale groups, and the impacts were higher than the full sample. This may be because the ability of small-scale farmers to manage cropland is weaker and the relative costs of cropland abandonment are smaller than those of medium- and large-scale farmers. For the household composition groups, a group of households without seniors gaining access to AMHSs significantly reduced the cropland abandonment, and the impacts were higher than the full sample. However, there were no significant impacts in the group of households with seniors. The above results also proved that those households with seniors were less likely to abandon their cropland, as the elderly in the households can manage the cropland [54]. For nonagricultural income, the AMHSs accessed by farmers significantly reduced cropland abandonment only in those households with nonagricultural income. These results verified Hypothesis 2, that is, that AMHSs can effectively alleviate the impact of labor migration on cropland abandonment. In addition, for different regions, the AMHSs accessed by farmers significantly reduced cropland abandonment only in the eastern region, with a higher level of economic development and more nonagricultural employment opportunities. This is consistent with Deng et al. (2018) [7], that is, the regions with a higher nonagricultural employment rate have more abandoned cropland.

**Table 4.** Heterogeneity analysis results.

| Variables | Different Cropland Scale Groups | | | Household Composition | | Nonagricultural Income | | Different Region | | |
|---|---|---|---|---|---|---|---|---|---|---|
| | Small-Scale | Medium-Scale | Large-Scale | With Seniors | Without Seniors | With Nonagricultural Income | Without Nonagricultural Income | Eastern | Central | Western |
| AMHSs access | −0.670 *** | −0.170 | 0.121 | 0.009 | −0.264 *** | −0.192 ** | −0.091 | −0.525 *** | −0.058 | −0.227 |
| | (0.135) | (0.132) | (0.147) | (0.149) | (0.086) | (0.077) | (0.287) | (0.174) | (0.101) | (0.168) |
| Control variables | Yes | Yes | Yes | Yes | Yes | Yes | Yes | Yes | Yes | Yes |
| Year dummy | Yes | Yes | Yes | Yes | Yes | Yes | Yes | Yes | Yes | Yes |
| Province dummies | Yes | Yes | Yes | Yes | Yes | Yes | Yes | Yes | Yes | Yes |
| Constant | −5.995 | −3.115 *** | −1.452 *** | −1.629 *** | −2.232 *** | −2.086 *** | −1.695 ** | −1.869 *** | −1.907 *** | −1.878 *** |
| | (296.596) | (0.446) | (0.350) | (0.378) | (0.238) | (0.204) | (0.738) | (0.496) | (0.304) | (0.279) |
| Instrumental variable | Yes | Yes | Yes | Yes | Yes | Yes | Yes | Yes | Yes | Yes |
| Endogenous test | 0.380 *** | 0.201 ** | −0.043 | 0.022 | 0.190 *** | 0.137 *** | 0.148 | 0.264 ** | 0.083 | 0.165 |
| | (0.084) | (0.094) | (0.092) | (0.101) | (0.058) | (0.053) | (0.185) | (0.114) | (0.070) | (0.105) |
| Wald $\chi^2$ | 323.30 *** | 245.40 *** | 206.50 *** | 178.03 *** | 473.53 *** | 547.13 *** | 52.87 *** | 72.84 *** | 312.43 *** | 341.16 *** |
| Observations | 2946 | 2702 | 2697 | 2628 | 5717 | 7345 | 1000 | 1496 | 3630 | 3219 |

*** $p < 0.01$, ** $p < 0.05$.

In addition, this study further analyzed the heterogeneity of household composition and nonagricultural income (i.e., with groups with nonagricultural income and containing seniors, with groups with nonagricultural income but without seniors). The results are shown in Table A1 (Appendix A), the AMHSs accessed by farmers significantly reduced cropland abandonment only in those groups of households with nonagricultural income but without seniors. This also proved that the rural elderly labor force can curb the impact of labor migration on cropland abandonment to a certain extent.

## 4. Discussion

Prior studies have proved that labor migration and high agricultural production costs are the main factors causing the abandonment of cropland [10,22]. Based on these factors and the actual situation of China's agricultural production, this study mainly focuses on the problem of whether AMHSs accessed by farmers can reduce cropland abandonment. We used the data from 8345 samples collected by the Survey for Agriculture and Village Economy in 2019 and 2020, and employed the extended probit regression model and the extended interval regression model to empirically analyze the relationship between AMHSs access and cropland abandonment. Our results revealed that AMHSs accessed by farmers can significantly reduce cropland abandonment in rural China. The research results can provide a theoretical reference for the government to promote agricultural mechanization services and reduce cropland abandonment.

Interestingly, the heterogeneity analysis results of our study showed that AMHSs accessed by farmers significantly reduced cropland abandonment in small-scale groups, groups of households without seniors, groups of households with nonagricultural income, those located in the eastern region, and groups of households with nonagricultural income but without seniors. On the one hand, household management by smallholders (smallholder refers to those who operate cropland area that is less than 3.33 hectares) is still the main form of agricultural management in China, and accounts for more than 98.00% of the total number of farmers [55]. They may not purchase agricultural machinery for agricultural production due to the small management scale, fragmented cropland, and high fixed costs of the agricultural machinery. In this case, the low level of agricultural mechanization, imperfect land transfer market, and high nonagricultural wages will make them more inclined to abandon their cropland, while accessing AMHSs can effectively reduce cropland abandonment. This finding is consistent with the Chinese government's policy aimed at promoting modern agricultural practices to small farmers through developing the agricultural mechanization services market. On the other hand, population aging is a social phenomenon faced by all countries in the world, including China with 13.50 percent elderly

people [56]. However, the surplus elderly labor force can still manage the cropland when the young and middle-aged labor force participate in non-agricultural work [26,54]. This undoubtedly proves that the elderly labor force may be an important resource to manage the cropland, which will reduce the cropland abandonment caused by labor migration. However, if the elderly labor force lacks farm successors, future land use issues should be a concern for scholars and governments, which may threaten food security and sustainable agriculture [57]. There are regional differences in the impacts of AMHSs access on cropland abandonment, which mainly has a significant impact on the eastern region. In addition, this study also proved that AMHSs access can alleviate the impact of labor migration on cropland abandonment.

This study provides some policy implications to reduce cropland abandonment. Our results show that AMHSs accessed by farmers can reduce cropland abandonment, which implies that policymakers should actively promote the development of agricultural mechanization services and build a perfect services market, especially for labor-intensive services (e.g., AMHSs). In addition, heterogeneity analysis showed that AMHSs have a more significant impact on reducing the abandonment of cropland by small-scale farmers, which also implies they may be the main group engaged in cropland abandonment in rural China. Thus, policymakers should strengthen the agricultural production support policies for small-scale farmers, such as subsidies for the use of agricultural mechanization services. This will help to realize the organic connection between small farmers and modern agriculture practice. Although elderly farmers can alleviate the cropland abandonment to some extent, the government should focus on farm successors in the future and continuously optimize the mode of agricultural mechanization services to better help farmers manage cropland.

This study mainly has two limitations as follows:

(1) This study only analyzed the impacts of AMHSs on cropland abandonment, while the impact of other services (such as agricultural machinery plowing, sowing, and irrigation services) was not analyzed. Although the AMHSs represent one of the "heaviest" agricultural production links. Thus, future research is required to explore the impact of other services on cropland abandonment, so as to provide a more comprehensive reference for the developing agricultural mechanization services and reducing cropland abandonment;

(2) Given the limitations of the paper length and questionnaire design, the potential channels (e.g., land transfer) of the impacts of AMHSs access on cropland abandonment have not been explored. Although we found that AMHSs can effectively alleviate the impact of labor migration on cropland abandonment, we still need to explore other channels in future research. In this case, we can provide more evidence on how to reduce cropland abandonment in rural China.

## 5. Conclusions

Based on the above analysis, the major conclusions are as follows:

(1) AMHSs accessed by farmers significantly reduced the probability of cropland abandonment by 18.5%;

(2) AMHSs accessed by farmers significantly reduced the proportion of the area of abandoned cropland in farmers' contracted cropland area by 20.3%;

(3) Heterogeneity analysis results showed that farmers' access to AMHSs significantly reduces cropland abandonment in small-scale groups, groups without elderly households, with nonagricultural income groups, in the eastern region, and in groups with nonagricultural income but without seniors.

In conclusion, our study confirms that AMHSs accessed by farmers can reduce cropland abandonment in rural China, which is also beneficial to ending hunger, achieving food security, and promoting sustainable agriculture.

**Author Contributions:** Conceptualization, P.X. and X.W.; methodology, P.X., Y.W. and X.H.; software, P.X.; validation, P.X., Y.W. and X.H.; formal analysis, P.X., X.H. and Y.W.; investigation, P.X., X.H. and X.W.; resources, P.X. and X.H.; data curation, P.X.; writing—original draft preparation, P.X. and X.W.; writing—review and editing, P.X., X.H., Y.W. and X.W.; visualization, P.X. and X.W.; supervision, X.W.; project administration, P.X. and X.W.; funding acquisition, X.W. All authors have read and agreed to the published version of the manuscript.

**Funding:** This research was funded by The Agricultural Science and Technology Innovation Program, grant number 10-IAED-08-2022, 10-IAED-RC-04-2022; and The National Key Research and Development Project, grant number 2020YFD1001205-1.

**Institutional Review Board Statement:** Not applicable.

**Informed Consent Statement:** Not applicable.

**Data Availability Statement:** The data presented in this study are available within the article.

**Conflicts of Interest:** The authors declare no conflict of interest.

## Appendix A

**Table A1.** Heterogeneity analysis results.

| Variables | Household Composition and Nonagricultural Income | |
|---|---|---|
| | **With Nonagricultural Income and Seniors** | **With Nonagricultural Income but without Seniors** |
| AMHSs access | −0.060 | −0.250 *** |
| | (0.158) | (0.091) |
| Control variables | Yes | Yes |
| Year dummy | Yes | Yes |
| Province dummies | Yes | Yes |
| Constant | −1.956 *** | −2.318 *** |
| | (0.481) | (0.249) |
| Instrumental variable | Yes | Yes |
| Endogenous test | 0.033 | 0.177 *** |
| | (0.109) | (0.062) |
| Wald $\chi 2$ | 172.70 *** | 421.71 *** |
| Observations | 2345 | 5000 |

*** $p < 0.01$.

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
