# Peer review of "Can Agricultural Machinery Harvesting Services Reduce Cropland Abandonment? Evidence from Rural China"

_agriculture, doi:10.3390/agriculture12070901_

Round 1

Reviewer 1 Report

Well done. The article is very well organized and the research data is also enough.

But I hope the author should add some environmental impacts of farmland abandonment in the introduction section, not just about food security.

Otherwise, please add a distribution map of the survey sample (a density map is also acceptable).

Also, I'm curious if there are geographical differences in your findings.

Reviewer 2 Report

Dear Authors,

The manuscript (agriculture-1770744) presented for review is interesting and important for the Chinese. I recommend minor corrections.

I propose to shorten the text of the manuscript because the authors repeat the same thing in several places. It seems that then the text would be more accessible to the reader.

Lines 109-111 – In my opinion, this text is not needed.

Figure 1 is not clear and should be improved.

Limitation: I propose to separate the limitations and show the strengths of these results.  

References: References are cited according to journal rules.

I believe it addresses an important area of research in an international context.

 Reviewer
